# The Effects and Mechanism of ATM Kinase Inhibitors in *Toxoplasma gondii*

**DOI:** 10.3390/ijms25136947

**Published:** 2024-06-25

**Authors:** Yangfei Xue, Zhu Ying, Fei Wang, Meng Yin, Yanqun Pei, Jing Liu, Qun Liu

**Affiliations:** 1National Animal Protozoa Laboratory, College of Veterinary Medicine, China Agricultural University, Beijing 100193, China; 17835422205@163.com (Y.X.); yzyingzhu@163.com (Z.Y.); wf_980614@163.com (F.W.); yinmeng9812@163.com (M.Y.); 18349325943@163.com (Y.P.); 2National Key Laboratory of Veterinary Public Health and Safety, College of Veterinary Medicine, China Agricultural University, Beijing 100193, China; 3Key Laboratory of Animal Epidemiology of the Ministry of Agriculture and Rural Affairs, College of Veterinary Medicine, China Agricultural University, Beijing 100193, China

**Keywords:** *Toxoplasma gondii*, ATM kinase inhibitors, cell division, protein secretion

## Abstract

*Toxoplasma gondii*, an important opportunistic pathogen, underscores the necessity of developing novel therapeutic drugs and identifying new drug targets. Our findings indicate that the half-maximal inhibitory concentrations (IC_50_) of KU60019 and CP466722 (abbreviated as KU and CP) against *T. gondii* are 0.522 μM and 0.702 μM, respectively, with selection indices (SI) of 68 and 10. Treatment with KU and CP affects the in vitro growth of *T. gondii*, inducing aberrant division in the daughter parasites. Transmission electron microscopy reveals that KU and CP prompt the anomalous division of *T. gondii*, accompanied by cellular enlargement, nuclear shrinkage, and an increased dense granule density, suggesting potential damage to parasite vesicle transport. Subsequent investigations unveil their ability to modulate the expression of certain secreted proteins and FAS II (type II fatty acid synthesis) in *T. gondii*, as well as including the dot-like aggregation of the autophagy-related protein ATG8 (autophagy-related protein 8), thereby expediting programmed death. Leveraging DARTS (drug affinity responsive target stability) in conjunction with 4D-Label-free quantitative proteomics technology, we identified seven target proteins binding to KU, implicated in pivotal biological processes such as the fatty acid metabolism, mitochondrial ATP transmission, microtubule formation, and Golgi proteins transport in *T. gondii*. Molecular docking predicts their good binding affinity. Furthermore, KU has a slight protective effect on mice infected with *T. gondii*. Elucidating the function of those target proteins and their mechanism of action with ATM kinase inhibitors may potentially enhance the treatment paradigm for toxoplasmosis.

## 1. Introduction

*Toxoplasma gondii* can induce severe complications and even mortality in individuals with compromised immune systems, including those with AIDS, organ transplant recipients, and cancer patients [1]. Currently, there is no efficacious drug regimen capable of addressing the diverse stages of the *T. gondii* life cycle [2,3]. Consequently, there is a pressing demand to develop novel therapeutic interventions for combating toxoplasmosis.

Parasites rely on complex absorption and synthesis mechanisms to fulfill their lipid requirements [4]. *T. gondii* employs distinct or non-existent lipid synthesis pathways (FAS) compared to its host. It contains three autonomous fatty acid synthesis pathways, including the FAS I pathway, that has not been fully elucidated for clearance, the FAS II pathway in the apicoplast, and the fatty acid extension pathway in the endoplasmic reticulum (ER) [5]. Since the FAS II pathway is not the primary source of fatty acids in mammalian cells, it has become a focal point in research for identifying specific drug targets against apicomplexan protozoa [5,6]. TgACBP1 (acyl-CoA-binding protein 1, ACBP1) plays an important role in the FAS II pathway and the lipid metabolism of *T. gondii* [7], while TgACBP2 (acyl-CoA-binding protein 2, ACBP2) is mainly involved in maintaining the mitochondrial function and cardiophospholipid metabolism in type II strains [5].To identify novel therapies for toxoplasmosis, we conducted virtual screening of drugs targeting the ACBPs proteins of *T. gondii* using an anti-obesity compound library (TOPSCIENCE, Shanghai) related to the fatty acid metabolism (Appendix A). Our investigation revealed that KU60019 (abbreviated as KU) significantly inhibits the in vitro growth of *T. gondii*. KU is an enhanced Ataxia telangiectasia mutated (ATM) protein kinase inhibitor [8], exhibiting an IC_50_ (half-maximal inhibitory concentrations) of 6.3 nM. ATM kinase inhibitors, including KU-559933, KU-60019, KU 59403, CP-466722 (abbreviated as CP), AZ31, AZ32, AZD0156, and AZD1390, have been assessed for their anti-tumor effects [9]. ATM serves as a core component of the DNA repair system, activated during DNA double-strand breaks to enhance homologous recombination (HR) repair pathways. Beyond its role in DNA damage repair, ATM governs cell cycle progression, apoptosis, chromatin remodeling, and transcription [10,11]. The deficiency of ATM kinase results in alterations in reactive oxygen species, mitochondrial dysfunction, transcription, an R-loop metabolism, and selective splicing, as well as defects in the cellular protein stability and metabolism [11]. *T. gondii* ATM (TgATM) kinase is indispensable for normal parasite replication [12], as evidenced by CRISPR screening analysis [13]. Due to the low similarity in the protein structure between TgATM and human ATM (HuATM) [12], we hypothesize that ATM kinase inhibitors may interact with multiple protein targets in *T. gondii*. Drug target identification technology based on drug affinity responsive target stability (DARTS) offers the advantage of not requiring the modification of small molecule compounds. Small molecule ligands bind to proteins and stabilize their conformational morphology, rendering them resistant to protease hydrolysis. By monitoring changes in the abundance of small-molecule protected proteins in protein samples treated with small molecules and proteases, the binding proteins of small molecules can be identified [14,15].

In this study, we have demonstrated the efficacy of ATM kinase inhibitors KU and CP in effectively suppressing the growth of *T. gondii* in vitro, leading to the abnormal division of daughter parasites. An examination under transmission electron microscopy revealed that treatment with KU and CP induced cellular enlargement, nuclear wrinkling, an uneven distribution of the plasma membrane, and increased dense granule formation in *T. gondii*, suggesting the potential impairment of parasite vesicle transport. Moreover, our investigations revealed their impact on the localization and expression of select secreted proteins, prompting the dot-like aggregation of the autophagy-related protein ATG8 (autophagy-related protein 8) in *T. gondii*, and hastening programmed cell death. Utilizing DARTS in conjunction with 4D-Label-free quantitative proteomics technology, we identified seven target proteins binding to KU, involved in numerous vital biological processes of *T. gondii*. Molecular docking studies further predicted their good binding affinity. Additionally, KU has a slight protective effect on mice infected with *T. gondii*.

## 2. Results

### 2.1. Evaluation of Anti-T. gondii Activity of ATM Kinase Inhibitors In Vitro

The effects of the ATM kinase inhibitors KU and CP on the growth of *T. gondii* were evaluated using the RH-luc (RH-luciferase) strain. After 72 h of infection, the expression level of luciferase was determined using a reagent kit. The results indicated a dose-dependent inhibition of *T. gondii* proliferation by KU and CP compared to the control group (Figure 1A,B), with IC_50_ values of 0.522 μM and 0.702 μM, respectively. Additionally, no significant cytotoxicity was observed within the range of treatment concentrations (Appendix A). The selection index (SI) was calculated to be 68 and 10 for KU and CP, respectively.

To further elucidate the key processes affected by KU and CP in the lytic cycle of *T. gondii*, RH-luc tachyzoites were treated with concentrations approximating IC_90_ and IC_50_. The results showed that both KU (Figure 2A,C) and CP (Figure 2B,C) significantly reduced the number and area of plaques formed by RH-luc tachyzoites. Additionally, it was observed that KU and CP had a relatively minor impact on the invasion of *T. gondii* (Figure 2D,E). However, at lower concentrations, they notably impeded the proliferation of *T. gondii* tachyzoites (Figure 2F,G). Treatment with KU and CP above 5 μM resulted in a predominant presence of *T. gondii* tachyzoites in a singular form, indicative of abnormal division. In summary, our preliminary findings suggest that KU and CP effectively inhibit the growth of *T. gondii* in vitro.

### 2.2. KU and CP Cause Abnormal Division of T. gondii Tachyzoites

The expression of inner membrane complex proteins (IMC1) is often closely related to the budding of daughter parasites [16]. The enoyl ACP reductase (ENR) protein can serve as a marker protein for apicoplast division [17]. To further confirm the effects of KU and CP on the division of *T. gondii*, RH tachyzoites were treated with the two compounds for 24 h, followed by IFA using anti-IMC1 and anti-ENR antibodies to label *T. gondii* replication. The results showed that both KU and CP induced asynchronous division and morphological abnormalities in *T. gondii* (Figure 3A), with 59% and 42% of observed abnormal division phenomena, respectively. Additionally, abnormalities in the division of the apicoplast of daughter parasites were observed, characterized by an odd number of apicoplasts (as indicated by the white arrow in Figure 3B). Specifically, 36% and 28% of apicoplast division abnormalities were observed in KU and CP, respectively. To visualize these abnormal divisions more clearly, a transmission electron microscope was utilized to examine the effects of the compounds on intracellular tachyzoites. The results demonstrated that the DMSO control group exhibited the normal division of tachyzoites and preserved the morphology of various organelles (Figure 3C). However, upon treatment with KU (Figure 3D) and CP (Figure 3E), *T. gondii* displayed asynchronous division, cell enlargement, nuclear shrinkage, an uneven distribution of the plasma membrane, and an increased density of granules (indicated by white arrows in Figure 3D,E). Additionally, the mitochondria were rarely observed, indicating impaired parasite vesicle transport and mitochondrial function. In summary, our findings suggest that KU and CP induce the abnormal morphology and division of *T. gondii* tachyzoites.

### 2.3. KU and CP Affect Protein Transport and Fatty Acid Metabolism in T. gondii

The glutathione S-transferase 2 (GST2) of *T. gondii* plays a crucial role in vesicular transport, and its absence results in a scattered point-like distribution of Rab5 (Rab-GTPase 5) and Rab6 (Rab-GTPase 6), as well as the decreased mislocalization and expression of several secreted proteins [18]. To further investigate the effect of compounds on the vesicular transport of *T. gondii*, we treated *T. gondii* with KU and CP, resulting in the diffuse localization of TgGST2 (Figure 4A). In addition, they also induced the abnormal localization of Rab5, leading to the chaotic localization of the early endosomes (Figure 4B). Compared with the control group, the compound treatment group resulted in the diffuse localization of GRA7 (Dense granule protein 7), MIC2 (Microneme protein 2), and ROP5 (Rhoptry protein 5) (Figure 4C,E and Figure 5C,E).

Due to the presence of KU in the anti-obesity-related small-molecule compound library, we next investigated whether it affects the fatty acid metabolism of *T. gondii*. The endogenous marker strains of ACBP1 and ACBP2 we constructed exhibited localization consistent with previous reports (Figure 5A,B and Appendix A). Treatment with KU and CP resulted in a significant decrease in the expression level of TgACBP1 (Figure 5C,E), while the expression level of TgACBP2 remained unaffected (Figure 5D,E). TgACBP1 participates in the metabolism of fatty acids, glycerides, and phospholipids, regulating parasite growth and pathogenicity [7]. In summary, our research findings suggest that KU and CP disrupt protein transport and fatty acid metabolism in *T. gondii*.

### 2.4. KU and CP Induced Autophagy and DNA Damage in T. gondii 

Previous studies have shown that the activation of ATM in cancer cells induces autophagy through the phosphorylation of BNIP3 (Interacting protein 3) [19]. Therefore, to verify whether the ATM kinase inhibitors KU and CP induce autophagy in *T. gondii*, we stained parasites with polyclonal antibodies against the autophagy-associated protein ATG8 and observed the dot-like aggregation of the ATG8 protein upon treatment with these compounds (Figure 6A). In addition, mammalian ATM kinases are involved in repairing DNA double-strand breaks [10]. To assess DNA damage in the compound-treated tachyzoites, TUNEL staining was performed on the RH-GFP parasite strain to detect DNA breaks. Approximately 56% and 42% of the parasites were TUNEL-positive, respectively (Figure 6B). When mammalian ATM is inhibited, the decrease in mitochondrial gene transcription activation leads to a decrease in mitochondrial biogenesis and a decrease in cellular ATP levels [11]. After treating tachyzoites with 5 μM KU and CP for 24 h, it was found that the ATP content of *T. gondii* significantly decreased (Figure 6C). HSP60 (heat shock protein 60, HSP60), as a mitochondrial marker protein, can usually assist in the correct transport, folding, and assembly of intracellular proteins [20]. KU and CP cause abnormal mitochondrial morphology in RHΔ*ku80* tachyzoites, presenting a spherical and fragmented morphology (Figure 6D). In addition, in the late stage of cell apoptosis, Caspase-Activated DNase (CAD) is activated by Caspase and cleaves the DNA between nucleosomes, forming DNA fragments in units of 180–200 bp. After agarose nucleic acid electrophoresis, DNA layers are formed [21]. After treatment with ATM kinase inhibitors KU, CP, AZD0156, and the positive control atovaquone for 24 h, DNA layers are observed in *T. gondii* (Figure 6E), as indicated by the white dashed box. These results indicate that KU and CP induce autophagy and DNA damage in *T. gondii*, thereby preventing cell cycle progression and inhibiting parasite proliferation.

### 2.5. The Effect of KU on the Transcription of T. gondii

Transcriptomics can provide new ideas for exploring potential therapeutic targets and drug development. The above research results suggest that KU affects multiple important biological processes of *T. gondii*. In order to explore the mechanism of KU’s action on *T. gondii* more comprehensively, we applied 5 μM KU to tachyzoites for 24 h and collected the parasites. We used transcriptomics to search for differentially expressed genes and determine the metabolic pathways affected. The criteria for screening differentially expressed genes were a *p*-value of less than 0.05 and a|log2 (foldchange)|greater than 1. The results showed that after 24 h of treatment with 5 μM KU in *T. gondii* tachyzoites, 224 expressed genes were upregulated and 203 expressed genes were downregulated (Figure 7A,B). KEGG (Kyoto Encyclopedia of Genes and Genomes) pathway enrichment analysis revealed that KU mainly affects metabolic pathways including sugar degradation, the pentose phosphorylation pathway, the amino acid and nucleotide metabolism, the sphingophospholipid metabolism, and carbon fixation in *T. gondii*, which is consistent with our previous research findings (Appendix A).

### 2.6. DARTS Combined with Proteomics to Determine the Binding Protein of KU in T. gondii

Considering the previous results confirming that KU affects multiple important biological processes of *T. gondii*, we suspected that there may also be multiple proteins that bind to it in *T. gondii*. Here, we used DARTS combined with 4D-label-free quantitative proteomics technology to identify seven proteins bound to KU (fold change > 2, *p*-value < 0.05, Figure 8A,B). According to the GO and KEGG enrichment analysis, their biological functions are mainly related to crucial processes such as fatty acid transport in *T. gondii*, mitochondrial ATP transport, microtubule formation, and the transport of Golgi proteins (Figure 8C,D).

Among the identified seven binding proteins, four proteins with known functions are TGGT1_273920 phenotypic aldose reductase (phenotype score, −1.59), TGGT1_293810 carboxyvinyl carboxylate phosphorylate phosphorylase (−0.6), TGGT1_220140 EF hand-domain-containing protein (−2.29), and TGGT1_297730 phenotypic transcription association factor 1 (−0.49). In addition, there are three putative proteins with unknown functions: TGGT1_246730 (−3.05), TGGT1_318490 (0.66), and TGGT1_236920 (−1.16). Docking simulation technology is a convenient and effective method to explore the interaction between small molecules and targets. Here, we employed molecular docking to study the docking of compounds KU and CP with seven differentially expressed proteins. The results showed that the seven differentially expressed proteins had good binding potential with KU and CP (Figure 9A–F), with binding affinity scores below −6 kcal/mol (Appendix A).

### 2.7. Evaluation Anti-T. gondii Activity of KU in Mice

To evaluate the clearance effect of KU on *T. gondii* in vivo, 8-week-old BALB/c mice (5 mice/group) were infected with 100 RH and 1000, 3000 Pru tachyzoites per mouse, respectively. A blank control group and a drug-only group were established and orally administered daily for one week. The results showed that 2 mice in the RH + Corn oil group died on the 8th day and all died on the 9th day. Two mice in the RH + 20 mg/kg KU group died on the 8th day and all died on the 10th day. One mouse in the RH + 40 mg/kg KU group died on days 9, 10, and 11, respectively, and all mice died on day 12. Two mice in the RH + 60 mg/kg KU group died on the 9th day, one died on the 10th day, and all died on the 11th day. This indicates that KU does not have a significant protective effect on mice infected with RH (Figure 10A). Among the 1000 Pru-infected mice, only the corn oil group had deaths by the 15th day, while the rest survived (Figure 10B). In the 3000 Pru-infected mice, two mice died in the corn oil group and one in the 20 mg/kg KU treatment group after two weeks of infection (Figure 10C). Compared with the Pru + Corn oil group, the weight loss of mice in the Pru + KU group was slowed down and the difference was significant (Figure 10D,E). After weighing the surviving mice 30 days later, they were euthanized, their brain tissue was removed and ground, and the number of brain cysts was counted. The results showed that compared with the Pru + Corn oil group, the number of brain cysts in the Pru + KU group mice was significantly reduced (Figure 10F,G). These results indicate that KU has a slight protective effect on mice infected with *T. gondii*.

## 3. Discussion

Due to the fact that the FAS II pathway is not the primary source of fatty acids in mammalian cells, it presents itself as a viable research avenue for identifying specific drug targets against *T. gondii* [5,6]. The initial high-throughput screening of small-molecule libraries, acquired from the company, computationally identified KU as a high-affinity binder to the acyl CoA-binding domain ACBD in ACBP1 and ACBP2 (with score values of −8.3 and −7.8, respectively). Upon treatment with these compounds, we observed a significant decrease in the expression level of TgACBP1, while the effect on the expression of TgACBP2 was minimal. This disparity may be attributed to the pivotal role of TgACBP1 in FAS II and the lipid metabolism of *T. gondii* [7], whereas TgACBP2 primarily contributes to mitochondrial function maintenance [5]. These findings suggest that ATM kinase inhibitors can indeed influence the fatty acid synthesis of *T. gondii*, but whether this influence is direct or indirect necessitates further investigation. It is also plausible that the treatment of *T. gondii* with KU and CP disrupts their protein transport and DNA replication, ultimately leading to the collapse of various metabolic processes.

KU, a specific ATM kinase inhibitor, has demonstrated efficacy against various tumor cells. In the lung cancer cell lines H1299 and A549, the combination of KU with VP-16 inhibits cell growth and survival while inducing higher rates of apoptosis [21], reflecting the accelerated programmed cell death observed in *T. gondii*. A deficiency in ATM kinase leads to reactive oxygen species (ROS) defects, mitochondrial dysfunction, compromised cellular protein stability, and an altered metabolism [13]. Additionally, an ATM deficiency reduces the efficiency of the pentose phosphate pathway (PPP) by decreasing the activity of glucose-6-phosphate dehydrogenase (G6PD) via the ATM-promoted phosphorylation of heat shock protein 27 (HSP27). This impairment likely contributes to the altered fatty acid metabolism observed in *T. gondii*, as lipid synthesis relies on glycolytic products as precursors [13]. In ATM-deficient cells, the reduced transcriptional activation of mitochondrial genes dependent on nuclear respiratory factor 1 (NRF1) leads to decreased mitochondrial biogenesis, providing an explanation for the rare observation of *T. gondii* mitochondria in transmission electron microscopy. Mitochondrial dysfunction ultimately leads to DNA damage and the excessive activation of poly (ADP-ribose) (PAR) polymerase (PARP) enzymes [13], resulting in protein aggregation, which may contribute to protein transport disruptions in *T. gondii*. TgGST2 is implicated in vesicular transport, and its absence leads to a scattered, punctate distribution of Rab5 and Rab6, as well as the reduced mislocalization and expression of several secreted proteins [18]. The treatment of *T. gondii* with KU and CP results in an increase in the dense granule observed under transmission electron microscopy, and further research is warranted to elucidate the specific protein transport pathways affected.

Since the homology between *T. gondii* ATM kinase and human ATM kinase is only 36%, the homology is mainly limited to kinase and FATC domains. The molecular weight of HuATM is 350–370 kDa, and the predicted molecular weight of TgATM is only 246 kDa, which is substantially different from HuATM [12]. Additionally, our research results indicate that KU and CP have an impact on multiple important biological processes of *T. gondii*, and we suspect that there are other binding proteins that interact with ATM kinase inhibitors in *T. gondii*. Target protein identification is crucial for identifying the mechanisms of action, side-effects, and evaluating the drug similarity of small-molecule drugs. The commonly used “labeling” targets require the synthesis of probes, which is time-consuming and may affect the conformation of the active drug [22,23]. Here, we used DARTS combined with differential proteomics to identify binding proteins and we identified seven binding proteins. According to GO (Gene Ontology) enrichment analysis, their biological functions are mainly related to important processes such as fatty acid transport, mitochondrial ATP transport, microtubule formation, and Golgi protein transport in *T. gondii*. This is consistent with the phenotype observed in compound treatment. Unfortunately, TgATM and TgACBP1 have not been identified, mainly due to the limitations of DARTS itself [24,25]. On the one hand, due to the complexity of the protein library, there may be some erroneous binding between small-molecule drugs and proteins; on the other hand, in SDS-PAGE and LC-MS analysis, positive results of some low-abundance proteins are easily overlooked. Furthermore, some erroneous operations may also lead to false-positive results, so it is necessary to verify the experimental results of DARTS through other experimental methods. In addition, it has been reported that HuATM recognizes over 900 regulated phosphorylation sites, including over 700 proteins [26]. The functions of these seven binding proteins and whether they are substrate proteins or have interactions with TgATM are also the focus of our next research. Our original intention was to investigate the function of TgATM (TGGT1_248530) and whether it interacts with KU; however, its genome is too large and its phenotypic value is too low to label or obtain knockout strains. Only a molecular prediction can prove their strong binding ability with KU and CP (scores −9.8 and −6.5). In addition, animal experiments have shown that KU has a slight protective effect on mice infected with *T. gondii*. The structural modification of KU may lead to drugs with better therapeutic effects on animal toxoplasmosis. In addition, AZD0156, an ATM inhibitor, exhibits high efficiency and specificity in inhibiting ATM kinases, as well as an excellent ability to penetrate the blood–brain barrier. This is currently being studied in phase I clinical trials, and follow-up research may also bring new strategies for the treatment of toxoplasmosis [9]. In summary, our study reveals that ATM inhibitors affect multiple important biological processes of *T. gondii*, and there are multiple binding targets.

## 4. Materials and Methods

### 4.1. Host Cells and Parasites

Human foreskin fibroblasts (HFF) and African green monkey kidney cells (Vero) were obtained from the ATCC Company in the United States and maintained in our laboratory. The endogenous marker strains constructed based on the RHΔ*Ku80* strain for this study were also preserved in our laboratory. The RHΔ*Ku80*, RH-luciferase (RH-luc), RH-GFP, and type II Pru strains were cultured in Vero/HFF cells supplemented with 1% fetal bovine serum (FBS) in Dulbecco’s modified Eagle’s medium (DMEM, M&C Gene, Beijing, China). The cultures were maintained in a 37 °C incubator with 5% CO_2_, with the culture medium refreshed every 24 h. The primers used are shown in Appendix A.

### 4.2. Construction of Transgenic Strains

To generate the TgPACBP1-HA and TgPACBP2-HA parasites, we first constructed the pLIC-HA-DHFR TgACBP1/2 plasmid, which allowed for the insertion of three HA tags at the C-terminus of the target genes within the genome locus. We designated the 42 bp fragment upstream of the gene translation termination codon as the 5’ flanking homologous arm, while the 42 bp fragment downstream of the gRNA served as the 3’ flanking homologous arm. Subsequently, we co-transfected linearized fragments along with corresponding CRISPR-Cas9 plasmids into the RHΔ*Ku80* parasite. Following transfection, parasites were subjected to screening with pyrimethamine to select for successful integration events. The design of gRNA was performed using E-CRISP (http://www.e-crisp.org/ECRISP/, accessed on 18 September 2023). The gRNA sequences used in this study were GAGGGCATTCCATCGCACGA and GAAGGTAATGGCCAGGAACG, respectively.

### 4.3. Drugs

The compounds KU60019 and CP466722 should be dissolved in dimethyl sulfoxide (DMSO) at their maximum solubility levels. Both DMSO and the small molecule compounds can be purchased from MedChemExpress (MCE, Shanghai, China) Biotechnology.

### 4.4. Cytotoxicity Assay

The cytotoxicity assay was performed as previously reported [27]. Cell viability was assessed using the Cell Count Kit-8 (CCK8, Beyotime, Shanghai, China) according to the manufacturer’s protocol. Initially, HFF cells were evenly seeded at a density of 1 × 10^4^ per well in a 96-well plate. After 4 h of incubation, the cells were treated with different concentrations of small molecule compounds and incubated further at 37 °C for 72 h. Subsequently, 100 μL of pre-mixed CCk8 solution (90 μL DMEM + 10 μL CCk8) was added to each well, followed by continued incubation for 2–4 h. 0.1% DMSO was used as the control well. Absorbance (OD450/OD630) was measured using an Epoch continuous wavelength enzyme-linked immunosorbent assay (ELISA) reader (BioTek Instruments, Winooski, VT, USA). The cell survival rate (%) was (experimental well reading-blank well reading)/(control well reading-blank well reading) × 100%. All experiments were performed in triplicate.

### 4.5. Luciferase-Based Growth Inhibition Assays

Luciferase assays was performed following previously reported procedures [28]. Inoculated 3 × 10^3^ RH-luc strains were added to HFF cells grown in 96-well plates containing or without compounds, resulting in a final well volume of 200 μL, with a DMSO content maintained below 0.1%. Following a 72 h incubation period, the luciferase activity was assessed using the firefly luciferase reporter gene detection kit (RG005, Beyotime, Shanghai, China). To perform the assay, the culture medium was discarded, and 100 μL of lysis buffer solution was added to each well, followed by incubation at 37 °C for 20 min to lyse the cells. Subsequently, 100 μL of luciferase substrate solution was added to each well, and the plate was shaken for 1 s. Luciferase activity was then measured using an Infinite 200 Pro multifunctional microplate reader (Tecan, Männedorf, Switzerland) after a 10 s period. Compound concentrations were calculated using GraphPad Prism software v8.0 (GraphPad, San Diego, CA, USA) based on the average of three biological replicates.

### 4.6. Plaque Assays

The plaque assays were performed as previously reported [29]. HFF cells were inoculated onto a 12-well cell plate, and upon reaching 90% confluence, newly released tachyzoites (100 per well) were inoculated along with compounds of different concentrations. To prevent concentration changes caused by evaporation, PBS was added to the wells in the outer ring. Following 7 days of incubation, the supernatant was discarded, and the cells were fixed with 4% paraformaldehyde and stained with crystal violet. The numbers and area of plaques were analyzed using GraphPad Prism software v8.0 (Adobe, San Jose, CA, USA) after repeating the experiment three times.

### 4.7. Anti-Invasion Assessment of Inhibitors In Vitro

The invasion assay was carried out as previously described [30]. Briefly, 1 × 10^5^ freshly released parasites were inoculated onto HFF cells cultured in 12-well plates containing different concentrations of compounds. The plates were then incubated at 37 °C for 40 min and subsequently washed with PBS. Fresh culture medium was added, and the plates were further incubated for 24 h. After the incubation period, the cells were fixed with 4% paraformaldehyde and stained with rabbit anti-GAP45 polyclonal antibody (diluted at 1:200) and Hoechst. The percentage of invasion was quantified by assessing the number of Parasitophorous Vacuoles (PVs) in each host cell. For each experiment, a minimum of 100 randomly observed PVs were counted. The assay was independently performed in triplicate. 

### 4.8. Anti-Proliferation Activity of Inhibitors

Freshly released tachyzoites (1 × 10^5^/well) were inoculated onto HFF cells. We allowed for 40 min of invasion time, followed by two washes with PBS. Subsequently, the media were replaced with culture media containing different concentrations of small molecule compounds. After 24 h, the cells were fixed with 4% paraformaldehyde and immunofluorescence staining was performed. We utilized a fluorescence microscope (Olympus Co., Tokyo, Japan) to observe and record the number of tachyzoites within 100 PVs. We repeated the experiment three times to ensure the reliability and consistency of results.

### 4.9. Immunofluorescence Assays and Western Blot

For the immunofluorescence assays (IFAs), HFF cells were infected with parasites for 24 h and then fixed with 4% paraformaldehyde for 20 min. Subsequently, the cells were permeabilized with 0.25% Triton X-100 for 20 min and blocked with 3% Bovine Serum Albumin (BSA) for 1 h. Following blocking, the cells were incubated with primary antibodies at 37 °C for 1 h. After washing three times with PBS, the cells were further incubated with secondary antibodies at 37 °C for another 1 h. Finally, images were captured using the Olympus IX70 inverted microscope (Olympus Co., Tokyo, Japan).

For the Western Blot (WB) assays, parasites were lysed using RIPA buffer (Beyond, Beijing, China) for subsequent Sodium Dodecyl Sulfate Polyacrylamide Gel Electrophoresis (SDS-PAGE). The separated proteins were then transferred onto a polyvinylidene fluoride (PVDF) membrane and blocked with 5% skim milk for 1 h. Following blocking, the membrane was incubated with a primary antibody at 37 °C for 1 h, washed, and then further incubated with a secondary antibody for 1 h before detection.

The primary antibodies used in this study comprised mouse-derived anti-HA Monoclonal Antibodies (Mab) purchased from Sigma (St. Louis, MI, USA). Additionally, mouse-derived anti-IMC1, anti-ENR, anti-ATG8, anti-ENR, anti-Actin, as well as anti-GAR7, anti-MIC2, and anti-ROP5, along with rabbit-derived anti-GAP45 and anti-HSP60 antibodies were sourced from our laboratory.

As for the secondary antibodies utilized, FITC-conjugated goat anti-mouse IgG (H + L) and goat anti-rabbit IgG (H + L), as well as Cy3-conjugated goat anti-rabbit IgG (H + L) and goat anti-mouse IgG (H + L) were acquired from Sigma (St. Louis, MO, USA). DAPI and Hoechst are commonly used for staining nuclear DNA. Furthermore, horseradish peroxidase (HRP)-conjugated secondary antibodies were employed for specific applications. 

### 4.10. Transmission Electron Microscope

Transmission electron microscopy (TEM) was performed as previously described [31]. In brief, tachyzoites treated with 5 μM KU and 5 μM CP for 24 h were subjected to gentle scraping to collect cells containing PVs using a cell scraper. After two washes, the samples were fixed with 3.5% glutaraldehyde at 4 °C for 48 h, followed by fixation in 1% osmium acid solution in darkness for 2–4 h. Subsequently, the samples were dehydrated using graded alcohols and acetone, and then embedded in epoxy resin for ultra-thin sectioning. Finally, the section was examined using transmission electron microscopy (RuliTEMHT7800, Hitachi, Japan) to visualize ultrastructural changes.

### 4.11. TUNEL Assay

The TUNEL assay was conducted following established procedures [32]. In brief, a one-step TUNEL detection kit (Beyotime Biotechnology, Shanghai, China) was utilized to assess DNA damage in *T. gondii*. RH-GFP tachyzoites treated with 5 μM KU and 5 μM CP for 24 h were collected for extracellular IFA. The parasites were fixed with 4% paraformaldehyde, permeabilized with 0.25% Triton X-100, and then subjected to labeling with the TUNEL reaction mixture for 1 h. The samples were observed under a fluorescence microscope.

### 4.12. ATP Content Detection

We used the instructions of the enhanced ATP detection kit (Beotime, S0027) to detect changes in the ATP content of *T. gondii*. In brief, fresh released RHΔ*ku80* tachyzoites were inoculated into fully fused HFF cells, cultured at 37 °C for 24 h, and then incubated with 5 μM KU and 5 μM CP treatment for 24 h. We collected intracellular tachyzoite and counted them, then lysed them on ice for 20 min, and centrifuged to obtain the supernatant for measurement. We established an ATP standard detection curve, prepared a detection working solution, added 100 μL ATP detection working solution to each well for 5 min, then added 20 μL prepared samples to the detection wells, shaken and mixed, measured the fluorescence intensity using an Infinite 200 Pro multifunctional microplate reader, and performed statistical analysis using GraphPad Prism software v8.0 (Adobe, San Jose, CA, USA).

### 4.13. Transcriptomics Analysis

We inoculated freshly released RHΔ*ku80* tachyzoites onto fully fused HFF cells, cultured at 37 °C for 24 h, and then treated with 5 μM KU for 24 h. We used cell scrapes and syringes to break the cells and release the parasite. We added 1 mL of Trizol (Sangon Biotech, Shanghai, China) to fix the sample. The control group was treated with 0.1% DMSO, and three replicates were set for each sample. Then, APPLIED PROTEIN TECHNOLOGY (Shanghai, China) was commissioned to conduct transcriptomic research and analysis.

### 4.14. DARTS and 4D-Label-Free Quantitative Proteomics

The experimental procedure for DARTS was conducted following established protocols [14,15]. Firstly, freshly released tachyzoites were collected and lysed using non-denatured protein lysates (M-PER, Thermo, Waltham, MA, USA) supplemented with protease and phosphatase inhibitors. The protein concentration was determined using a BCA assay kit. Subsequently, the experimental group was treated with 100 μM KU, followed by incubation on ice for 1 h and then further incubation at 37 °C for 20 min. Pronase E (MCE, Shanghai, China) was added to the samples at a ratio of 1:100, and hydrolysis was performed at room temperature for 15 min. The hydrolysis reaction was stopped by adding a protease inhibitor cocktail and cooling the samples on ice. For the quantitative proteomic analysis, unlabeled Liquid Chromatography-Mass Spectrometry (LC-MS) was conducted with three parallel samples, as per technical guidance provided by APPLIED PROTEIN TECHNOLOGY (Shanghai, China). Subsequently, the obtained data were compared with the TOXODB database search results, and MaxQuant software (version 1.6.1.0) was utilized for further analysis and interpretation.

### 4.15. Molecular Docking

The three-dimensional structure of the proteins employed in docking was predicted using the Swiss Model online website. Subsequently, the Sitemap module in Maestro 13.0 was utilized to forecast the active pocket of each protein. The structures of KU and CP were obtained from the PUBCHEM database. Molecular docking simulations were conducted using AutoDock Vina 1.1.2 software. The docking results were analyzed, and the highest scoring docking conformation was deemed as the binding conformation. PyMol 2.5.4 was employed for a visual analysis of the docking results. The Uniprot IDs of TGGT1_273920, TGGT1_293810, TGGT1_220140, TGGT1_297730, TGGT1_246730, TGGT1_318490, and TGGT1_236920 are S7V3L0, Q2L8W6, S7UGW6, A0A125YVV4, S7W9L9, A0A125YG85, and A0A125YZE1, respectively.

### 4.16. Anti-T. gondii Activity of KU in Mice

A total of 100 freshly released RH Δ*Ku80* tachyzoites per mouse and either 1000, 3000 Pru tachyzoites per mouse were used to infect 8-week-old BALB/c mice via an intraperitoneal injection. After 24 h of infection, mice were subjected to treatment with doses of 20, 40, and 60 mg/kg of KU, which were diluted in corn oil. The mice were treated with oral gavage administered once daily for 7 consecutive days. Throughout the experiment, the survival rate and weight changes of the mice were monitored, and their weight was recorded every other day.

### 4.17. Statistical Analysis

Data were analyzed using GraphPad Prism software v8.0 (Adobe, San Jose, CA, USA). All statistical analyses were conducted using the two-tailed Student’s *t*-test. The results of the survival curves were analyzed using the Log-rank (Mantel-Cox) test. In the figures, the *p* values are represented by asterisks as follows: * for *p* < 0.05, ** for *p* < 0.01, and *** for *p* < 0.001, while “ns” indicates no statistically significant difference. For this study, a significance level of *p* < 0.05 was considered statistically significant.

## 5. Conclusions

ATM kinase inhibitors KU and CP effectively suppress the growth of *T. gondii* and induce aberrant division. KU and CP disrupt protein transport and trigger autophagy in *T. gondii*, hastening programmed cell death. Additionally, KU has a slight protective effect on mice infected with *T. gondii*. Using DARTS in conjunction with 4D-label-free quantitative proteomics technology, we identified seven target proteins binding to KU.

## Figures and Tables

**Figure 1 ijms-25-06947-f001:**
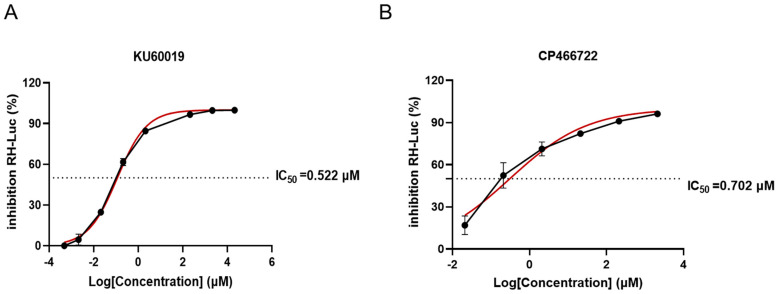
KU and CP inhibit the growth of *T. gondii* in vitro in a dose-dependent manner. (**A**) Growth inhibition curve of RH-luc treated with KU for 72 h. (**B**) Growth inhibition curve of RH-luc treated with CP for 72 h. IC_50_ values were calculated using the log (inhibitor) vs. normalized response variable slope in GraphPad Prism software v8.0 (San Diego, CA, USA) based on the concentration gradient set by CC_50_. Means ± SD of three repeated experiments, nonlinear regression-curve fitting.

**Figure 2 ijms-25-06947-f002:**
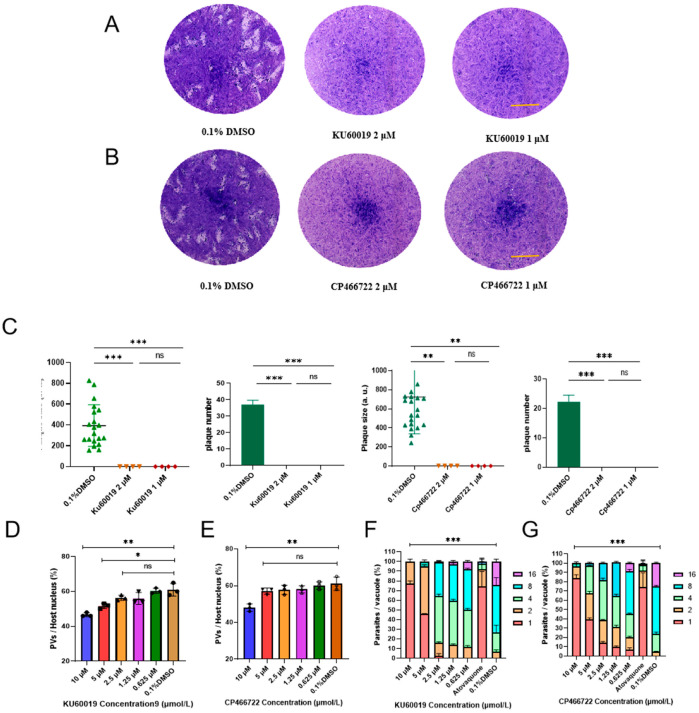
Effect of KU and CP on the proliferation of *T. gondii*. (**A**,**B**) Plaque formation ability and statistical analysis of *T. gondii* treated with KU. Scale bar, 1 cm. (**C**) Plaque formation ability and statistical analysis of *T. gondii* treated with CP. Statistical analysis of plaques area and size was conducted using Adobe Photoshop CS6 software (Adobe, San Jose, CA, USA). (**D**,**E**) Impact of KU and CP on the invasion of *T. gondii*. Invasion rate is expressed as the ratio of the number of parasitophorous vacuoles (PVs) to the nucleus in different fields of view. (**F**,**G**) Effects of KU and CP on the proliferation of *T. gondii*, quantified by the number of parasites within 100 PVs. The results were plotted using GraphPad Prism software v8.0 (GraphPad, San Diego, CA, USA) and presented as mean ± standard deviation. In all images, * *p* < 0.05, ** *p* < 0.01, *** *p* < 0.001, ns, no statistically significant difference.

**Figure 3 ijms-25-06947-f003:**
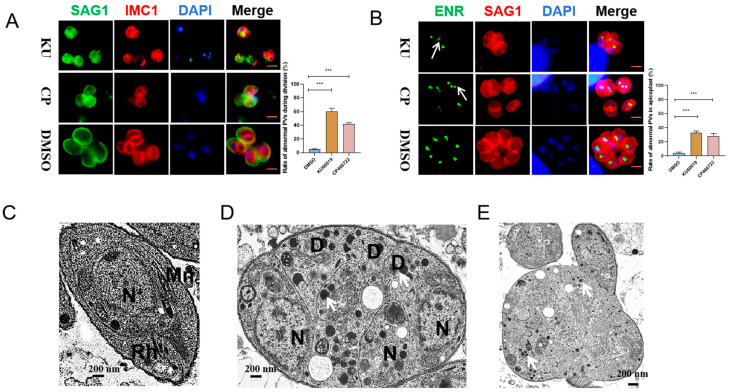
KU and CP induce abnormal division of *T. gondii*. (**A**,**B**) Effects of KU and CP on *T. gondii* division, visualized by IFA using IMC1 and ENR. Tachyzoites treated with 5 μM KU and CP for 24 h were labeled with IMC1 antibodies to visualize daughter parasites, while SAG1 antibodies were used to label the parasites’ plasma membrane. The white arrow indicates an odd number of apicoplasts. Abnormal division was quantified using GraphPad Prism software v8.0 (GraphPad, San Diego, CA, USA) based on counting in 100 PVs. Scale bar = 2 μm. *** *p* < 0.001. (**C**–**E**). Fixed sections of intracellular parasites were observed under a transmission electron microscope after 24 h of treatment with KU and CP. (**C**) DMSO control group showed normal division of tachyzoites. (**D**,**E**) Treatment groups with KU an CP exhibiting abnormal morphology and an increase in dense granules, as indicated by the arrows. Scale bar = 200 nm. Rh, Rhoptries; D, Dense granules; Mn, mitochondrion; N, Nucleus.

**Figure 4 ijms-25-06947-f004:**
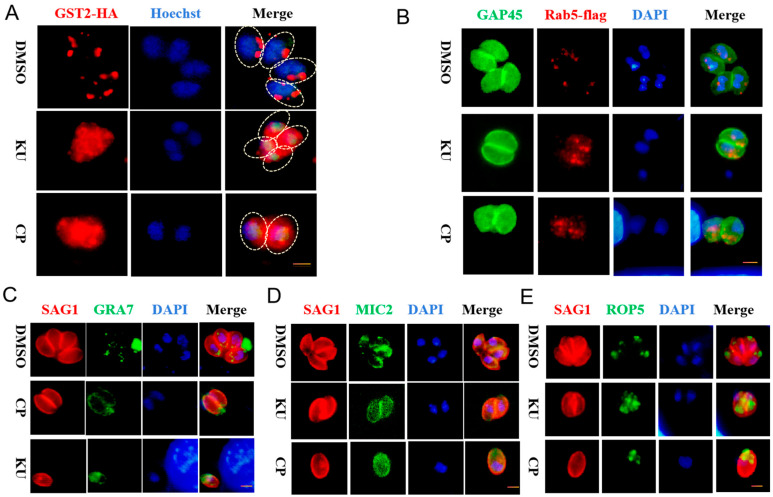
Effect of KU and CP on protein transport in *T. gondii*. (**A**) IFA showing the impact of KU and CP on GST2-HA (red) localization. (**B**–**E**) Localization of Rab5-flag (red), GRA7, MIC2, and ROP5 (green) following 24 h of treatment with the two compounds. Parasite morphology was visualized using anti TgSAG1 (red) and TgGAP45 (green), with the nucleus stained blue with Hoechst or DAPI. Scale bar = 2 μm.

**Figure 5 ijms-25-06947-f005:**
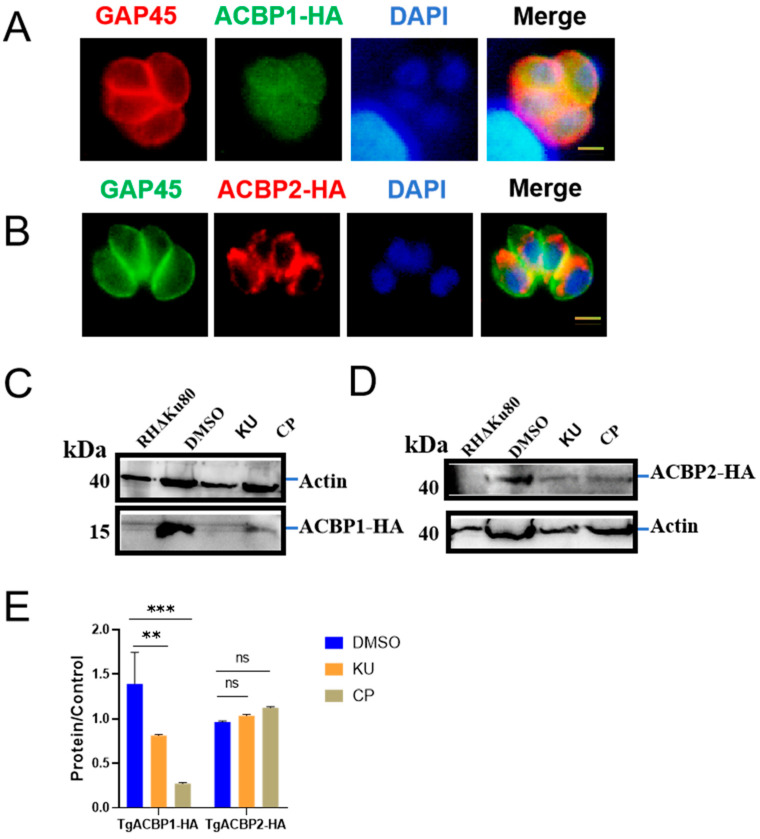
Effects of KU and CP on metabolism of fatty acids of *T. gondii*. (**A**) Localization of TgACBP1 in the cytoplasm of parasites. (**B**) Localization of TgACBP2 in parasite mitochondria. (**C**–**E**) Parasites were treated with 5 μM KU and CP for 24 h, followed by Western blot (WB) analysis to detect changes in the expression levels of ACBP1 and ACBP2. Differences in protein bands were analyzed using mouse-derived HA monoclonal antibody. ** *p* < 0.01, *** *p* < 0.001, ns, no statistically significant difference.

**Figure 6 ijms-25-06947-f006:**
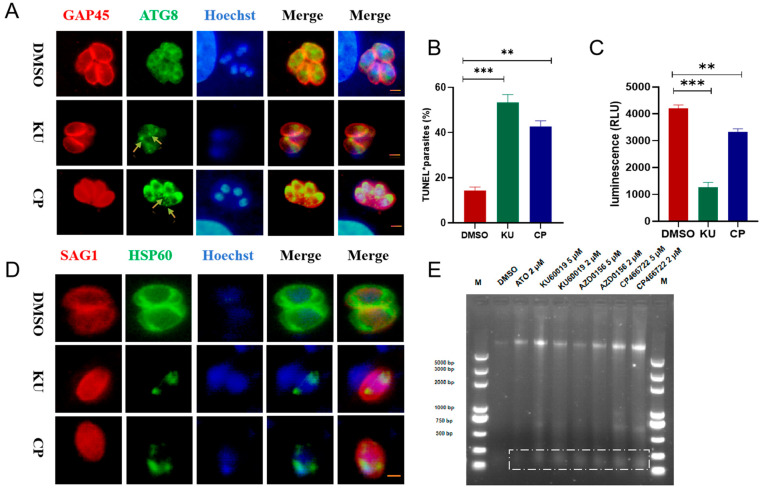
Two compounds induce autophagy and DNA damage in parasites. (**A**) IFA detected the localization of autophagy-related protein TgATG8 (green) after treatment with KU and CP. Antibodies used were polyclonal antibodies derived from mice against ATG8 and rabbits against GAP45. The yellow arrow indicates the dot like aggregation of ATG8 protein. (**B**) Fresh tachyzoites were collected for extracellular IFA after 24 h of treatment with the compounds, with GFP and TUNEL staining. The probability of DNA damage in 100 PVs was calculated. ** *p* < 0.01, *** *p* < 0.001. (**C**) An enhanced ATP detection kit was used to detect changes in ATP content in RHΔ*ku80* tachyzoites treated with 5 μM KU and CP for 24 h, with the vertical axis representing relative fluorescence intensity. ** *p* < 0.01, *** *p* < 0.001. (**D**) IFA detected the localization changes of TgHSP60 protein in intracellular parasites treated with 5 μM KU and CP for 24 h. The primary antibodies were rabbit-derived HSP60 (green) and mouse-derived SAG1 polyclonal antibody (red). Scale bar = 2 μm. (**E**) After treating RHΔ*ku80* tachyzoites with different concentrations of compounds for 24 h, the whole genome of the parasite was extracted for nucleic acid electrophoresis, with DMSO as the negative control and 2 μM Atovaquone as the positive control. ATO: Atovaquone.

**Figure 7 ijms-25-06947-f007:**
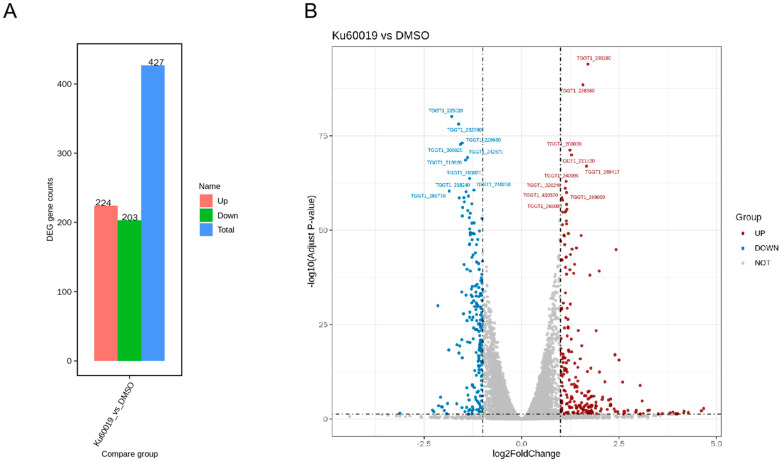
The effect of KU on the transcription of *T. gondii*. (**A**) The statistical results of differentially expressed genes are presented in the form of a bar chart. (**B**) The distribution of differential gene expression is visualized through volcano plots. The horizontal axis represents the difference in gene expression multiples between samples, while the vertical axis reflects the statistical significance indicator of gene expression level changes—*p*-value. Each point in the graph corresponds to a gene, with gray dots indicating genes that did not show significant differences, red dots indicating genes with significantly increased expression levels, and blue dots representing genes with significantly decreased expression levels.

**Figure 8 ijms-25-06947-f008:**
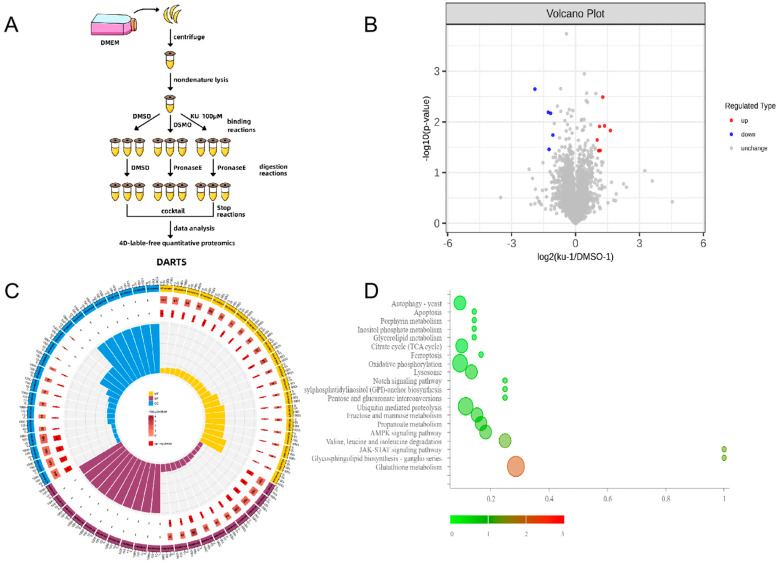
Identification of KU-binding protein in *T. gondii* using proteomics. (**A**) DARTS experimental flowchart. The enzymatic concentration of PronaseE is 1:100 and the enzymatic hydrolysis time is 15 min. (**B**) Volcano plots showing log2 protein rates vs. −log2 *p* values of global proteins in samples treated with KU and hydrolyzed with protease compared to samples without adducts but hydrolyzed with protease. Each sample has three replicates. (**C**) GO analysis of proteins with differential expressions (Appendix A). (**D**) KEGG analysis of proteins with differential expressions.

**Figure 9 ijms-25-06947-f009:**
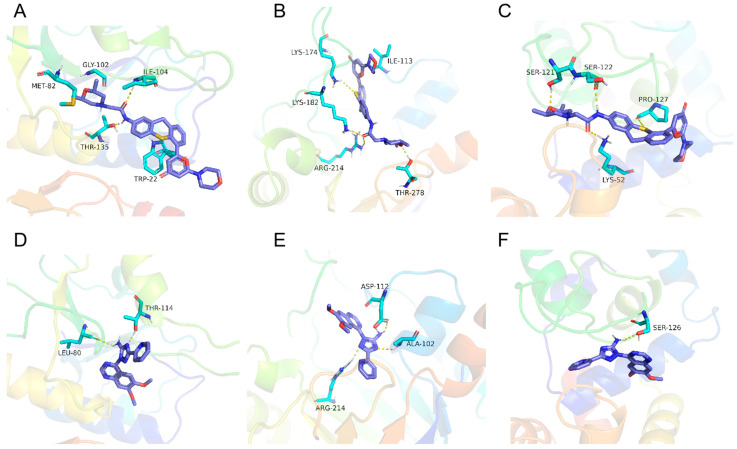
Binding patterns of KU and CP with differentially expressed proteins. (**A**–**C**) Local interaction views between KU and TGGT1-318490, TGGT1-293810, and TGGT1-22140 proteins, respectively. The docking scores are −9.5, −9.2, and −8.3 kcal/mol. (**D**–**F**) Local interaction views between CP and TGGT1-318490, TGGT1-293810, and TGGT1-22140 proteins, respectively. The docking scores are −7.8, −7.6, and −7.4 kcal/mol. The blue stick represents small molecules, the light blue cartoon represents binding amino acids, and the yellow dashed line represents hydrogen bonding interactions.

**Figure 10 ijms-25-06947-f010:**
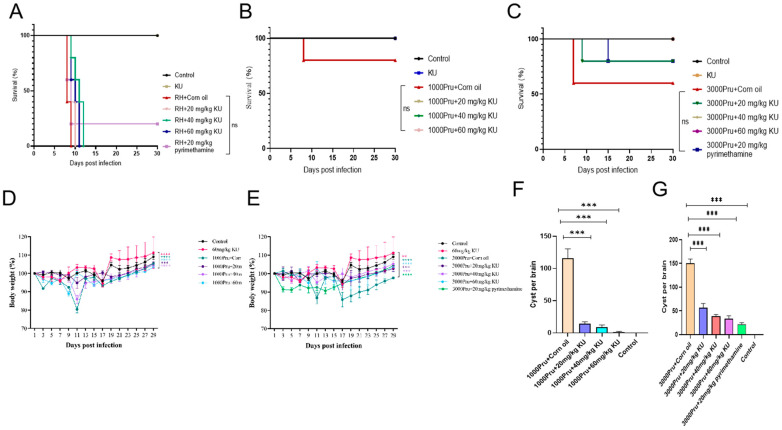
KU has a moderate protective effect on mice infected with *T. gondii*. (**A**) Survival curves of 100 RH-infected mice treated with different doses of KU. ns, no statistically significant difference. (**B**) Survival curves of 1000 Pru-infected mice treated with KU. ns, no statistically significant difference. (**C**) Survival curves of 3000 Pru-infected mice treated with KU. ns, no statistically significant difference. (**D**) Weight changes of 1000 Pru-infected mice after 30 days. *** *p* < 0.001, **** *p* < 0.0001. (**E**) Thirty-day weight changes in 3000 Pru-infected mice. ** *p* < 0.01, *** *p* < 0.001, **** *p* < 0.0001. (**F**) Statistics on the number of brain cysts in surviving mice infected with 1000 Pru. *** *p* < 0.001. (**G**) Statistics on the number of brain cysts in surviving mice infected with 3000 Pru. *** *p* < 0.001.

## Data Availability

All datasets generated for this study are included in the manuscript and its Appendix A.

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
