# Peer review of "The Effects and Mechanism of ATM Kinase Inhibitors in Toxoplasma gondii"

_ijms, 2024, doi:10.3390/ijms25136947_

Round 1

Reviewer 1 Report

Comments and Suggestions for Authors

In this study by Xur et. al. the authors have proposed compounds (KU60019 and CP466722) against T. gondii. They test the compounds in silico, in vitro and in vivo (mice). They find that application of these compounds leads to abnormal morphology and division in T. gondii. Additionally, they were able to link the compounds to effects on protein transport, fatty acid metabolism, transcription, DNA damage and autophagy in T. gondii. They also performed computational studies using molecular docking and homology modeling of proteins. I appreciate the authors for using different tools to provide a holistic view of why they think these compounds work and how they work. I find this work worthy of a follow up to optimize these compounds to see if they hold sufficient therapeutic relevance. Overall, the manuscript is well written and easy to read. I have few comments to share.

The Manuscript shows 22% similarity with other works on the web. This is bit high for my comfort. While I understand some parts of Material and methods could be falsely accused of plagiarism. But Line 42: “It harbors three autonomous pathways for fatty… the uncharacterized FASI pathway” is directly taken from another publication. Authors should avoid such direct copy of statements from other authors even if they acknowledge the source.

Line 26 (Line 85,272-275 and others) in Abstract: “Molecular docking predicts their extremely high binding affinity.” I would like to object to use of the words “extremely high”. I accept that the predicted binding affinity is good, but it is not “extremely high”. This is misleading.

I observe that the proteins used in the computational part were modelled. I would like the authors to give more details on how they were built (more than Swiss Model was used). Details on what Uniprot ID was used, which template(s) were used should be described to enable replication of results by manuscript readers. Authors should also perform some tests to show that the models they built are robust and reliable. Minor point: while Swiss Model is a great tool and I myself have used it many times in the past, but I think state of the art modeling tool these days seems to be AlphaFold. I hope both Swiss Model and AlphaFold give same results for these proteins.

As the protein structures built were models and not experimentally determined I would have preferred the manuscript to have a short Molecular Dynamics simulation of some of the most relevant protein-ligand complexes to infuse more confidence in stability of both modelled protein structure and protein-ligand association. I acknowledge that Molecular Dynamics might not be authors forte in which case they should at least provide results from multiple docking tools to show consensus.

Provide Uniprot ID of proteins used in this study to enable easy connection to more information available online. Uniprot IDs are connected to many primary and secondary databases along with literature sources.

Increase font size of labels in Figure 9.

I would have greatly appreciated if authors could have done some optimization of these compounds to propose better compounds. But I understand that this might be out of scope for this manuscript.

Author Response

Dear Reviewers

Thank you so much for the thoughtful feedback from the reviewers on our manuscript “The effects and mechanism of ATM kinase inhibitors in Toxoplasma gondii”. We appreciate the time and effort of the 2 reviewers to providing constructive comments. Those comments are all valuable and very helpful for revising and improving our manuscript, as well as the important guiding significance to our researches. We have studied comments carefully and have made correction which we hope meet with approval. Please find below our point-by-point responses to each of the reviewer’s comments.

Reviewer 2 Report

Comments and Suggestions for Authors

This manuscript by Xue et al. tested the effect of ATM kinase inhibitors against Toxoplasma gondii. Although the compounds have good in vitro anti-Toxoplasma efficacy with nanomolar IC50s, their efficacy in the acute toxoplasmosis murine model is poor. Both compounds seem to target multiple parasite proteins, and further studies delineating their mechanism of action would be interesting. While the science is technically robust, the experimental section seems incomplete, with some missing sections.

The manuscript requires a moderate revision before it can be accepted for publication. I have some comments and suggestions:

1.     Introduction, Line 47: Please write out the complete form of ACBP on first use, followed by the abbreviation in parentheses.

2.     Introduction, Lines 49-51: The authors should consider omitting the definition of virtual drug screening and the accompanying references, as this information is well-known to the journal's reader base.

3.     Results & Methods: Information regarding the high-throughput screening of the small molecule library is missing from the results and methods section. Please incorporate this in the revised manuscript.

4.     Results, Line 95, Figure S1: Strangely, the initial cell viability starts from over 100% and falls below 100% in Figures C and D, respectively. Did the authors use any controls for the cytotoxicity experiments? Also, how was the cell viability calculated?

5.     Results, Lines 105-109, Figure 2: The references to Figure 2 are mostly incorrect. For example, Figure 2H does not exist in the actual figure. Please fix the figure, figure legend, and text in the Results section.

6.     Results, Line 140, Figure 3E: White arrows indicating the increased density of dense granules in treated parasites are not visible in Figure 3E.

7.     Results, Line 175, Figure 5: The references to Figure 5 are incorrect. For example, Figure 5C shows the expression level of TgACBP2, while Figure 5D shows the expression level of TgACBP1.

8.     Results, Line 207: Please correct the spelling of atovaquone. Also, it would be helpful for the readers to indicate DNA layers in Figure 6E by arrows.

9.     Results, Lines 253, 256: The authors seem to want to reference Figure 8 instead of Figure 7 here.

10.  Results, Lines 274: It seems that the authors want to reference Figure 9A-F instead of Figure 8A-F here.

11.  Results, Line 288: Please indicate the average survival time to compare the two groups. The treatment does not seem to significantly enhance the survival time of the RH-infected mice.

12.  Results, Line 289: Did the authors quantify the brain cysts in the Pru-infected mice? Type II Toxoplasma strains like the Pru do not have a high mortality rate at infection doses used by the authors, as also shown by the results obtained in this study. Therefore, comparing treatment efficacies based on monitoring survival rates is useless in such infections. Comparing brain cyst yields would be a much better approach for testing the efficacy of the treatments.

13.  Methods: Please provide the gRNA sequences used in the study.

14.  Methods, Line 403: Please recheck the formula for cytotoxicity quantification.

15.  Methods, Line 409: What kind of plates were used for luciferase assays?

16.  Results, Figure 2, Methods: The authors mention that the invasion rate was expressed as the ratio of the number of parasitophorous vacuoles (PVs) to the nucleus in different fields of view; however, the y-axis label on the figure indicates parasites and not PVs.

17.  Methods, Sections 4.8, 4.12, & 4.13: Change the tense to past tense.

18.  Methods, Line 532: Were mice treated orally? Please mention this in the text.

19.  Discussion & Conclusions: Based on the results from the animal studies, KU seems to have, at best, a minor effect against toxoplasmosis. Please replace the text in the manuscript accordingly to reflect this.

Comments on the Quality of English Language

While the quality of the English language is good overall, a few sections need changing the tense.

Author Response

(The authors gave the same response as above.)
